# Salicylic Acid Mediates Chitosan-Induced Immune Responses and Growth Enhancement in Barley

**DOI:** 10.3390/ijms252413244

**Published:** 2024-12-10

**Authors:** Pawel Poznanski, Abdullah Shalmani, Marcin Bryla, Waclaw Orczyk

**Affiliations:** 1Plant Breeding and Acclimatization Institute—National Research Institute, Radzików, 05-870 Blonie, Poland; p.poznanski@ihar.edu.pl (P.P.);; 2Institute of Agricultural and Food Biotechnology—State Research Institute, Rakowiecka 36, 02-532 Warsaw, Poland; marcin.bryla@ibprs.pl

**Keywords:** barley transcriptome, cereal crops, deacetylation degree, *Fusarium graminearum*, *Hordeum vulgare*, necrotic reactions, pathogenesis related (PR) proteins, *Puccinia hordei*, reactive oxygen species (ROS), systemic acquired resistance (SAR)

## Abstract

Chitosan (CS), derived from the partial deacetylation and hydrolysis of chitin, varies in the degree of deacetylation, molecular weight, and origin, influencing its biological effects, including antifungal properties. In plants, CS triggers immune responses and stimulates biomass growth. Previously, we found that the antifungal activity of CS was strongly dependent on its physicochemical properties. This study revealed that the chitosan batch CS_10 with the strongest antifungal activity also effectively activated plant immune responses and promoted biomass growth. Barley treated with CS_10 exhibited systemic acquired resistance (SAR), characterized by micronecrotic reactions upon *Puccinia hordei* (*Ph*) inoculation and reduced symptoms following *Fusarium graminearum* (*Fg*) infection, representing biotrophic and necrotrophic pathogens, respectively. CS_10 treatment (concentration 200 ppm) also enhanced plant biomass growth (by 11% to 15%) and promoted the accumulation of salicylic acid (SA), a hormone that regulates both plant immune responses and growth. Low levels of exogenous SA applied to plants mirrored the stimulation observed with CS_10 treatment, suggesting SA as a key regulator of CS_10-induced responses. Transcriptomic analysis identified SA-regulated genes as drivers of enhanced immunity and biomass stimulation. Thus, CS_10 not only fortifies plant defenses against pathogens like *Ph* and *Fg* but also boosts growth through SA-dependent pathways.

## 1. Introduction

Chitin, the second most abundant natural polysaccharide after cellulose, is a key component of crustacean and insect exoskeletons, fungal cell walls, and cephalopod beaks. It is a linear polymer composed of β-1,4-linked N-acetyl-D-glucosamine (GlcNAc) [1]. As the main component of fungal cell walls, chitin oligomers released during pathogenesis act as plant immune elicitors. Chitosan (CS), derived from the partial deacetylation and hydrolysis of chitin, is a copolymer of N-acetyl-D-glucosamine (GlcNAc) and deacetylated D-glucosamine (GlcN). Each CS batch varies in its degree of deacetylation (DD), average molecular weight (MW), and biological origin [2]. CS is soluble in water at acidic pH (at pH < 6), when most amino groups are protonated. Depending on their molecular characteristics, the CS molecules can influence biological processes in diverse organisms with strong antifungal properties that restricts the growth of pathogenic fungi [3] and significantly reduce mycotoxin levels in plant material [4,5]. Chitosan serves as an effective elicitor of plant immunity, enhancing plants’ defenses against a wide range of pathogens, including bacteria, fungi, and viruses. In addition to its protective role, chitosan application promotes plant growth, boosts yield, and stimulates the production of diverse secondary metabolites [6]. These beneficial characteristics, along with biocompatibility and biodegradation, make CS a suitable candidate for various applications, including sustainable agriculture [7,8,9,10]. Its strong antifungal effects against *Fusarium* species, inhibition of mycotoxin accumulation [4], and enhancement of plant immunity position makes CS a promising compound for mitigating *Fusarium*-related diseases. However, the precise biological mechanisms by which CS affects both fungal and plant processes are only partially understood and warrant further investigation.

Barley (*Hordeum vulgare* L.) is a major cereal crop globally, but it is highly susceptible to Fusarium head blight (FHB), a severe disease affecting various cereals. FHB is primarily caused by the *Fusarium oxysporum* species complex (FOSC), a soil-borne pathogen that includes *Fusarium graminearum* (*Fg*). FOSC ranks fifth among the top economically damaging pathogens, impacting over 100 crop species. FHB not only reduces crop yields but also degrades grain quality due to the accumulation of harmful mycotoxins, which can render the harvest unsuitable for food and feed use [11,12,13]. *Fg* exhibits a biotrophic growth phase early in infection, followed by a necrotrophic phase during active pathogenesis, during which it produces mycotoxins, including deoxynivalenol (DON). The Golden Promise cultivar was selected for this study due to its widespread use as a model in cereal genetics and breeding research. It is particularly valuable because its entire genome has been sequenced and fully annotated, providing a comprehensive resource for genetic and functional studies.

Plant immunity activated by pathogen-associated molecular patterns (PAMPs) involves the reprogramming of host cellular processes to resist pathogen attacks. Chitin- and chitosan-derived elicitors serve as fungal PAMPs, triggering defense responses in plants [14]. These responses include the rapid accumulation of reactive oxygen species (ROS) at the infection site, which act as signaling molecules that activate the plant immune response and local necrotic reactions. ROS-dependent necroses directly impact pathogens, with varying effects based on their life strategies and pathogenesis types. For instance, biotrophic pathogens, like *Ph*, are quickly and strongly countered by ROS and local micronecrotic reactions [15,16,17]. In contrast, necrotrophic pathogens such as *Fg* can exploit host necrosis to enhance their pathogenesis. However, low to moderate ROS levels serve as signaling molecules that activate additional defense mechanisms in both pathogenesis systems. The plant immune system is highly sophisticated, with pathogen recognition being the critical initial step in preventing disease progression [18].

Salicylic acid (SA) is a plant hormone that regulates immune responses by controlling numerous crucial genes involved in defense [19,20]. Beyond its role in immunity, SA is also involved in stress response [21] and growth regulation [22,23]. Plants synthesize SA through two main pathways: the ICS (isochorismate synthase)-dependent and the PAL (phenylalanine ammonia-lyase)-dependent pathways [20,24]. The predominance of one pathway over the other varies by species and is influenced by environmental or biotrophic factors; in some cases, both pathways can operate simultaneously. For instance, rice primarily relies on the PAL-dependent pathway [25], while in Arabidopsis, the ICS-dependent pathway is dominant [26]. In soybeans, both PAL- and ICS-dependent pathways are equally important [27]. The involvement of each pathway can vary depending on the type of immune response or environmental condition. In barley, there are seven PAL paralogs and a single ICS gene [28], but their roles in SA biosynthesis are still under debate. Hao et al. [29] reported that the single ICS gene plays a major role in SA biosynthesis and the development of SA-dependent basal resistance to *Fg*. In their system, none of the known PAL paralogs influenced SA levels or resistance to *Fg*. Conversely, Qin et al. [30] found that knocking out the ICS gene did not affect basal SA levels, suggesting that the PAL-dependent pathway is responsible for maintaining them. These findings suggest that SA biosynthesis in barley is complex, likely involving both pathways. The PAL-dependent pathway may maintain basal SA levels, while ICS-dependent synthesis is triggered by external signals. In a previous article [4], we explored the antifungal effect of CS application, focusing on its ability to mitigate pathogenesis and inhibit mycotoxin synthesis. We found that only a specific batch of chitosan_10 (CS_10), characterized by a low molecular weight (MW) of 30 kDa and a high degree of deacetylation (DD) of ≥90, exhibited the desired antifungal properties in a concentration of 200 ppm [4]. CS oligomers showed no significant antifungal activity, while high MW CS batches were relatively ineffective.

In the current experiments, we discovered that the CS_10 sample with the strongest antifungal properties was among those which most effectively stimulated barley growth and immune responses. Therefore, the objective of this article was to investigate *Hordeum vulgare* cv Golden Promise response to CS_10 application, aiming to identify the factors responsible for: (i) enhanced immune response and resistance and (ii) increased plant biomass growth.

## 2. Results

### 2.1. CS_10 Treatment Activates Plant Immune Response in Barley Leaves Detectable as Smaller Necrotic Symptoms and Stronger Micronecroses After Fg and Ph Inoculations

Addressing the first hypothesis that CS_10 treatment activates the immune response, the two plant pathogens, which represented two different lifestyles, i.e., necrotrophic *Fg* and biotrophic *Ph*, were used. The *Fg* infection symptoms on the third leaves of mock-treated control plants showed medium to large dark-colored necrotic areas. The large chlorotic sections indicated widespread infection. The infection symptoms on the third leaves of CS_10-treated plants were smaller compared to the control. The dark necrotic spots were restricted to the inoculation sites, and they were surrounded by relatively small chlorotic areas. In mock-treated plants, necrotic sites and chlorotic areas were visibly larger compared to the control (Figure 1A). The rate of *Fg* infection in both variants was further quantified by qPCR using primers designed for *Fg (Fg_TRI5*) and barley (*Hv_EFG1*) genes (Appendix A) and gDNA from the infected leaves as a template. In mock-treated plants, the ratio of *Fg*-specific *TRI5* gene per one copy of barley-specific *EFG1* varied from 0.11 to 0.21, with the average value of 0.15 (SD ± 0.04). In CS_10-treated plants, the ratio of *Fg_TRI5*/*Hv_EFG1* varied from 0.03 to 0.09, with an average of 0.06 (SD ± 0.02) (Figure 1B).

Six days post-inoculation with *Ph*, the third leaves of the CS_10-sprayed plants exhibited fewer necrotic spots and weaker infection symptoms compared to the mock-treated plants (Figure 2A). Calcofluor white (CW) staining followed by the microscopic visualization of the infection sites showed the development of *Ph* infection structures and the presence of micronecrotic reactions of plant cells on different time days (Figure 2B).

The germination of *Ph* urediniospores was observed already at 1-day post-inoculation (dpi) in mock-treated control, while in the CS_10-treated plants, a similar stage was observed a day later at 2 dpi. The subsequent stages of *Ph* development were delayed by about one day in CS_10-treated plants compared to the control (Figure 2B). The first micronecrotic reactions representing plant response, which further restricted *Ph* pathogenesis, were observed 2 dpi in CS_10-treated plants and 3 dpi in the control plants (Figure 2B). The CW-stained samples allowed for the quantitative scoring of infection sites with micronecroses. In mock-treated control plants, micronecroses were not detectable at 1 and 2 dpi. The infection rate of micronecroses at 3 dpi was 7.8%, and it increased to 11.5% and 15.8% at 4 and 5 dpi, respectively. In CS_10-treated plants, the micronecrotic responses were observed one day earlier compared to mock at 2 dpi, and the rate was 4.8%. On subsequent days, the rates were 17.7, 26.2, and 36.2% at 3, 4, and 5 dpi, respectively (Figure 2C). Moreover, the samples of *Ph*-inoculated leaves collected from mock- and CS_10-treated plants were stained with DAB followed by microscopic observations. Brown precipitate, the result of DAB oxidization by hydrogen peroxide, indicated sites of hydrogen peroxide accumulation in plant tissues. The intensity of the DAB staining also allowed for the semi-quantitative analysis of H_2_O_2_ accumulation. A higher accumulation of H_2_O_2_ was observed in the CS_10-treated plants during the early days of infection, particularly at 3 and 4 dpi. In contrast, H_2_O_2_ accumulation was higher in the mock-treated plants after 5 dpi. Additionally, we noted that the number of infections was higher on the third leaf of the mock-treated plants compared to the CS_10-treated plants, which may be due to the early accumulation of H_2_O_2_ in the CS_10-treated plants (Figure 2D).

### 2.2. CS_10 Treatment of Barley Plants Leads to Elevated Levels of Salicylic Acid in the Leaves

SA concentration was quantified in the pooled second and third leaves of 14-day-old plants. The leaves were mock treated (*Hv*_mock), sprayed with CS_10 200 ppm solution (*Hv*_CS), and inoculated with *Fg* macrospores (*Hv*_*Fg*). The SA concentration were measured for all treatments in two timepoints: one day after treatment and three days after treatment. One day after mock treatment, the level of SA in mock-treated, CS_10-treated, and *Fg*-inoculated plants were 975 µg/kg, 967 µg/kg, and 1471 µg/kg, respectively. Three days after treatment, SA concentrations in mock-treated, CS_10-treated, and *Fg*-inoculated plants were 785 µg/kg, 1298 µg/kg, and 1646 µg/kg, respectively (Figure 3).

### 2.3. Treatments with CS_10 and SA Stimulates the Biomass Growth of Barley Seedlings

To investigate the stimulation of plant growth by different samples of CS (Table 1), barley seedlings were grown in a semi-hydroponic system in Hoagland medium. First, we investigated how different CS samples (Table 1), characterized by different deacetylation degrees (DDs), molecular weights (MWs), and biological origins, affected the growth of barley seedlings (Figure 4A). The relative biomass gains of the barley seedlings treated with different CS samples ranged from 0.93 to 1.16. Treatment with CS_5 and CS_100-300 showed a slight inhibition of growth by factors 0.93 and 0.96, respectively. Treatment with CS_8-15, CS_10, CS_10-120, CS_30-100, and CS_300-1k enhanced biomass growth by factors of 1.16, 1.11, 1.07, 1.06, and 1.13, respectively (Figure 4A). Although biomass enhancement was relatively modest, it was statistically significant for CS samples CS_8-15, CS-10, CS_10-20, and CS_300-1k. For subsequent experiments, CS_10 was chosen due to its superior antifungal activity against *Fg*.

In another set of experiments, the relative biomass gain after treatment with CS_10 200 ppm was 1.15 (SD ± 0.17) (Figure 4B). The relative biomass gain after treatment with SA 50 µM was 1.10 (SD ± 0.21) and SA 400 µM was 1.00 ± 0.20) (Figure 4B).

### 2.4. Transcriptomic Analysis of Barley Response to CS_10 Treatment and Fg Inoculation

Barley transcriptome was analyzed in the following variants: barley inoculated with *Fg* (*Hv*_*Fg*), treated with CS_10 and inoculated with *Fg* (*Hv*_*Fg*_CS), barley treated with CS_10 (*Hv*_CS) and the control, mock-treated (*Hv*_mock) plants used as reference (Figure 5).

Initial variations between different variants were analyzed using hierarchical clustering distance (Pearson average distance on top 3000 genes) (Figure 6A). The correlation matrix represents the global relation between all variants. The average correlation coefficients (R^2^) of gene expression between biological replicates of each treatment were high, ranging between 0.94 and 1. Treatment wise, *Hv*_mock and *Hv*_CS were closest, followed by *Hv*_*Fg*_CS and *Hv*_*Fg* (Figure 6B). All 12 samples (3 for each of the 4 variants) were analyzed with the use of principal component analysis (PCA). The treatments were separated by the first principal component (PC1), which accounted for 82.5% of the variation. The biological replicates grouped closest were based on their respective treatment (Figure 6C). Globally, the biggest difference was observed between *Fg*-inoculated (*Hv*_*Fg* and *Hv*_*Fg*_CS) and *Fg*-non-inoculated samples (*Hv*_mock and *Hv*_CS), indicating that inoculation with *Fg* had a significantly stronger effect than treatment with CS_10 (Figure 6).

Genes with a false discovery rate (FDR) < 0.05 and genes with log2fold change > 2 were considered as differentially expressed genes (DEGs) between the variants (*Hv*_CS, *Hv*_*Fg*, and *Hv*_*Fg*_CS) in relation to the control (*Hv*_mock). The highest number of DEGs, 6652 downregulated and 6736 upregulated genes, was found in plants inoculated with *Fg* (*Hv*_*Fg*). *Fg* inoculation and CS_10 treatment (*Hv*_*Fg*_CS) led to 4715 upregulated and 3796 downregulated genes. CS_10 treatment (*Hv*_CS) led to downregulated 514 genes and upregulated 1198 genes (Figure 7A). The CS_10-treated sample (*Hv*_CS) showed 122 unique upregulated and 182 downregulated genes. An additional 91 upregulated and 39 downregulated genes were commonly found in the CS_10 treatment (*Hv*_CS) and CS_10 treatment and *Fg* inoculation (*Hv*_*Fg*_CS) variants. In the group of total 1198 genes upregulated after CS_10 treatment (*Hv*_CS), 985 (972 + 13) genes were also upregulated after *Fg* inoculation (*Hv*_*Fg*), and out of 514 downregulated genes, 293 (279 + 14) of these genes also overlapped with *Hv*_*Fg* downregulated genes (Figure 7B).

### 2.5. The Top Five Terms of Biological Processes (BPs), Molecular Functions (MFs), and Cellular Components (CCs) Are More Strongly Affected by CS_10 Treatment than Fg Inoculation

Gene set enrichment analysis (GSEA) provides information on pathway enrichment across analyzed samples. It allows researchers to identify enriched or depleted sets of genes and describe them by respective Gene Ontology terms. This approach gives an overview of the main barley transcriptome responses differing in CS_10 treatment and *Fg* inoculation (Figure 8). In the two tested variants (*Hv*_CS and *Hv*_*Fg*), the biggest fold enrichment of biological processes (BPs) was in the “L-phenylalanine catabolic process”. Similarly, “Aromatic amino acid family catabolic process” and “L-phenylalanine metabolic process” were among the most upregulated.

In *Hv*_CS, the top five downregulated GO terms were associated with the cell wall, lignin metabolic process, and plant cell wall organization. Differently, in *Hv*_*Fg*, the terms related to chloroplasts and photosynthesis were downregulated. In molecular functions (MFs), “Ammonia-lyase activity” and “Chitinase activity” were upregulated in both variants, *Hv*_CS and *Hv*_*Fg*. Large, downregulated differences between the variants were related to oxidoreductase activities. In cellular components (CCs), the upregulated terms in *Hv*_CS were related to cell wall, plasma membrane, plasmodesma, and cell–cell junction. In *Hv_Fg*, the terms related to proton transport, and extracellular space were upregulated, while the terms related to photosynthesis were downregulated (Figure 8).

### 2.6. CS_10 Treatment and Inoculation with Fg Activates PAL-Dependent SA Synthesis and Strong Upregulation of PR-Encoding Immunity-Related Genes

SA functions as an important regulator of immune responses. Here, the genes related to SA synthesis and the SA-dependent signaling pathway were manually selected from the pool of genes with FDR > 0.05 and marked green if upregulated and red downregulated. This approach indicates a set of genes up- or downregulated in association with a particular variant. It is worth noting that out of the two PAL- and ICS-dependent SA biosynthesis pathways, the genes participating in the PAL-dependent pathway (*PAL1*, *PAL2*, *PAL3*) were upregulated in all variants, while the *ICS* gene was downregulated in all variants (Figure 9). The NPR1 protein functions as the main regulator of SA-dependent signaling pathways. In CS_10-treated variants, the NPR1-encoding gene was upregulated, while in the variant of only *Fg* inoculation, the expression of *NPR1* was on the same level as the control (Figure 9).

*NPR1* was upregulated in CS_10-treated barley (*Hv*_CS and *Hv*_*Fg*_CS), while no change in expression was observed in barley inoculated only with *Fg* (*Hv*_*Fg*). Among other genes considered as receptors and regulators of the SA-mediated immune response, *NPR3* was upregulated across all treatments. In contrast, *NPR4* showed increased expression in *Fg*-inoculated barley (*Hv_Fg*) and CS_10-treated barley (*Hv*_CS) but not in barley simultaneously treated with both *Fg* and CS_10 (*Hv_Fg*_CS) (Figure 10). The expression of *NPR1*-activated genes encoding pathogenesis-related (PR) proteins displayed similar patterns of up- and downregulation across treatments. *PR1* to *PR5* genes had higher expression in *Hv_Fg* and *Hv_Fg*_CS than in *Hv*_CS. For example, *PR1* was highly expressed in *Hv_Fg* (7.9) and *Hv_Fg*_CS (6.5) compared to *Hv*_CS (2.1). Conversely, *PR6*, *PR7*, and *PR14* showed negative or low expression in *Hv_Fg* and *Hv_Fg*_CS, while they had moderate to low expression in *Hv*_CS. Genes such as *PR9*, *PR10*, *PR15*, and *PR16* were more highly expressed in *Hv_Fg* and *Hv_Fg*_CS than in *Hv*_CS (Figure 10).

Selected WRKY transcription factors (*WRKY6*, *WRKY33*, *WRKY70*) were consistently upregulated across all variants (Figure 10). *WRKY6* showed the highest expression in the *Fg*-inoculated and CS_10-treated variant *(Hv_Fg*_CS) (3.8) compared to only CS_10-treated (*Hv*_CS) (2.5) and only *Fg*-inoculated barley (*Hv_Fg*) (2.2). In contrast, *WRKY33* had the highest expression in the *Fg*-inoculated variant (*Hv_Fg*) (3.9), compared to the CS_10-treated (*Hv*_CS) and *Fg*-inoculated barley (*Hv_Fg*). *WRKY70* showed consistent, moderate upregulation across all treatments (Figure 10). Genes’ names and IDs are shown in Appendix A.

### 2.7. RT-qPCR Analysis of Selected Barley Genes Confirms the Reliability of RNA-Seq Data

The four genes, *NPR1*, *PR9*, *PR4*, and *PR14*, each with a known role in plant immune response, were selected and used for the RT-qPCR verification of RNA-seq analysis. The results of RT-qPCR transcript quantification of the four genes were highly correlated (R^2^ = 0.9329) with RNA-seq quantification (Figure 11). The results confirm the reliability of the RNA-seq results.

## 3. Discussion

Pathogens remain the main threat to global food security despite advancement in resistance breeding and the widespread use of agrochemicals. One reason for this persistent threat is the rapid evolution of new virulent strains of pathogens, which overcomes both genetic resistance and the active ingredients of plant protection chemicals. Natural antifungal agents, like CS, tend to target multiple and complex pathways, as opposed to agrochemicals, which typically focus on single well-defined targets. This broader targeting reduces the likelihood of pathogens developing resistance to natural compounds. Although CS’s antifungal activity is relatively modest, it can, when combined with partial genetic resistance, limit the progress of pathogenesis and potentially prevent field epidemics. In our previous research, we identified CS_10 as a batch of CS with significant antifungal activity, particularly in restricting *Fg* growth [4]. Using the same batch of CS_10, we treated barley plants to examine its effect on plant immune response, specifically looking for its ability to enhance resistance against pathogens with different modes of pathogenesis: the necrotrophic *Fg* and the biotrophic *Ph*. In our experimental setup, CS_10 was applied to the second leaves, while pathogen inoculation was conducted on the third leaves. This spatial separation allowed us to focus on the systemic immune response induced by CS_10 in plants rather than the direct antifungal action of CS_10.

The results showed that the CS_10 treatment reduced symptoms of *Fg* infection in the third leaves of plants treated with CS_10 compared to the mock-treated control (Figure 1B). This was confirmed by quantifying *Fg* gDNA in infected leaves. Plants treated with CS_10 exhibited a significant lower ration of *Fg* gDNA to barley gDNA—less than half of the level observed in mock-treated plants (Figure 1B). In barley (cv. Golden Promise), which is susceptible to the *Ph* strain used in the experiments, we measured micronecrotic reactions in response to *Ph* infection. This type of response is a hallmark of partial resistance to biotrophic *Puccinia* sp. first described by Niks [31]. It slows the rate of disease and restricts the field of epidemics [32]. In mock-treated plants, micronecrotic reactions were first detected at 3 days post-inoculation (dpi) in 7.8% of infection sites, increasing to 17.8% by 5 dpi. In CS_10-treated plants, this reaction occurred one day earlier, at 2 dpi, and reached a ratio of 36.2% by 5 dpi (Figure 2). The results confirmed that CS_10 treatment triggered a systemic immune response in the leaves adjacent to those directly treated with CS_10, resulting in a faster and more widespread immune activation. The necroses induced in response to *Ph* infection in CS_10-treated plants resembled cell death patterns typically seen in prehaustorial non-host resistance or adult plant resistance against *Puccinia* sp. pathogens in barley and other *Triticeae* crops [33,34,35]. Both sets of experiments, i.e., with *Fg* and *Ph*, provided results that are consistent with those previously reported on CS_10-triggered immunity (CTI) against *Fg* in chickpea [36]. The authors observed coordinated metabolic and physiological changes after the application of CS_10, including the fortification of the extra-cellular matrix (ECM), elevated ROS, and ROS-dependent signaling. Li, et al. [37] reported a similar range of systemic acquired resistance (SAR)-like responses after CS_10 priming, followed by *Fusarium zanthoxyli* inoculation of prickly ash. It included the upregulation of twelve key genes related to SAR, increased levels of H_2_O_2_, elevated activities of peroxidase and catalase enzymes, and enhanced accumulation of lignin and flavonoids. In all plants, a significant reduction of the lesions from 46.8% to 75.1% was observed [38]. Elsharkawy, et al. [39] found the same set of physiological changes in wheat plants treated with CS_10 and inoculated with *Puccinia* urediniospores. This mode of response in which the CS_10-primed plant shows an oxidative burst and elevated activities of redox enzymes is the prerequisite of micronecroses and remains in line with the results presented in this manuscript.

We can conclude that CS_10 treatment activates a systemic-like immune response, which, in turn, delays pathogenesis and lowers the infection symptoms of *Fg* or, in another *Ph* pathosystem, initiates a faster and more widespread micronecrotic response.

Further analysis revealed elevated levels of salicylic acid (SA) in *Fg*-inoculated and CS_10-treated plants. SA accumulation occurred as early as 1-day post-inoculation with *Fg*, while CS_10 treatment induced a similar rise 3 days after treatment (Figure 3). The results confirmed that SA synthesis and accumulation were part of the plant response to both factors in the pathogen inoculation and the application of CS_10 with the former perceived by host cells as a fungal elicitor. *Fg* infection, as a stronger biotic stressor, activated SA accumulation on the first day of pathogenesis compared to the CS_10 treatment, which activated a very similar response two days later. The results are consistent with current knowledge on SA-dependent regulation in response to molecular elicitors of biotic stressors. In plant cells, SA can be synthesized through either the ICS- or PAL-dependent pathways, depending on the plant species. For example, the ICS-dependent SA synthesis was reported in Arabidopsis [40] and in barley [29], while PAL-dependent SA synthesis was found in rice [25,41]. In soybean, the cooperative PAL- and ICS-dependent SA synthesis was reported [27]. The authors highlighted the importance of PAL in the pathogen-induced synthesis of SA. Furthermore, they found that pathogen infection suppressed the expression of the ICS-encoding gene. In our system, only PAL-encoding genes were upregulated in CS_10-treated and *Fg*-inoculated plants. The levels of upregulation differed in the tested variants, indicating that all three paralogs shared their roles and participated in the SA biosynthesis pathway. Conversely, the ICS-encoding gene was repressed in all variants tested (Figure 9), mirroring similar findings in soybean inoculated with pathogens [27]. In tomatoes treated with CS_10, as reported by van Aubel, et al. [42], the genes regulated by SA-dependent pathways were activated and enhanced immunity against powdery mildew. The authors found a significant upregulation of pathogenesis-related (PR) proteins and SA-related genes.

The results gathered in our system showed that SA was involved in response to CS_10 treatment and *Fg* inoculation, and the PAL-dependent pathway is the predominant route for SA synthesis in response to CS_10 and *Fg*.

Although numerous studies have reported growth stimulation following CS_10 treatment, systemic investigations under controlled conditions are rare. In our study, we expected that CS_10-induced biomass gains would be relatively small and that the gains would be masked by biomass variation between individual plants. Considering this, we adopted a semi-hydroponic system, which allowed us to track the biomass gains of individual plants and compared it with a mock-treated control. This approach confirmed that under controlled conditions, CS_10 treatment stimulated growth, and the relative biomass gain was 1.15 vs. 1.0 of the control. It should be noted that, despite a relatively small gain value, the significance of the results is high because of the large number of plants tested in five independent biological replications. Parallel experiments were performed to test whether SA treatment would affect plant growth. The results indicated that the SA application (50 µM) stimulated plant growth, and the relative gain in biomass was 1.10 vs. 1.00 of the control. The higher SA concentration (400 µM) had no effect. SA is a key regulator of immune reactions and plant growth, and the final outcome, i.e., growth stimulation or suppression, depends on SA concentrations and differs in different plant species. Shakirova, et al. [43] reported that 50 µM SA stimulated the growth of wheat seedlings by increasing the mitotic index in the root meristem and cell enlargement in the root extension zone. A similar effect was reported in Arabidopsis and *Matricaria chamomilla*. In Arabidopsis, Pasternak, et al. [44] showed that 50 µM SA treatment stimulated adventitious root formation and affected the apical meristem while concentrations greater than 50 µM inhibited the processes. The authors, in elucidating the mechanism, found that exogenous SA treatment led to changes in auxin synthesis and transport. In *Matricaria chamomilla*, 50 µM SA promoted growth, while 250 µM delayed it [45]. Generally, for a particular plant species, lower SA doses showed a stimulating effect, while larger doses restricted growth [46]. In our system, CS_10 treatment led to SA increase and simultaneously stimulated plant growth. We propose that plant growth stimulation observed after CS_10 treatment is dependent on SA.

The RNA sequencing of leaves treated with CS_10, inoculated with *Fg*, or treated with both revealed that *Fg* inoculation had the most significant impact on gene expression (Figure 7). Gene Ontology (GO) terms sorted by fold enrichment revealed that the top biological processes (BPs) were ‘L-phenylalanine catabolic process’, ‘L-phenylalanine metabolic process’, and ‘Systemic acquired resistance (SAR)’ (Figure 8). The first two categories are mutually related due to the PAL-dependent synthesis of SA, and this hormone is the main regulator of immune reactions, including SAR, which represents the third category. The most enriched categories of molecular functions (MFs) included ‘Ammonia-lyase activity’, ‘Chitinase activity’, and ‘phenylalanine lyase activity’ (Figure 8). They also represented functions directly related to PAL-dependent metabolic pathways, SA biosynthesis, and SA-regulated immune response. The cumulative analysis of the variants tested (Figure 9) indicated that the PAL-dependent pathway is the active pathway in our experimental system and each of the three PAL paralogs had its share in SA biosynthesis. Notably, *NPR1* (*NON-EXPRESSOR OF PR GENES1*), a key regulator of SA-mediated SAR, showed diverse expression patterns, confirming its active role in barley’s immune response to both CS_10 and *Fg* inoculation (Figure 9).

The transduction of SA-dependent signaling is coordinated by *NPR1* and leads to the activation of genes encoding pathogenesis-related (PR) proteins. The transcriptomic results showed the upregulation of ten PR-encoding genes, the downregulation of one, and a mixed regulation pattern of two other genes (Figure 10). *PR3*, *PR8*, and *PR11* represent diverse classes of chitinases, which hydrolyze chitin, the structural component of the fungal cell wall, and restrict fungal pathogenesis [47]. In our system, *PR3* showed the strongest upregulation in all variants tested (Figure 10), and this may account for restricted symptoms of *Fg* pathogenesis. Along with *PR3* and *PR2*, the (1-3)-β-glucanase encoding gene showed very strong upregulation. Both proteins have antifungal activities that effectively restrict *Fusarium* pathogenesis in wheat, as reported by [48]. In another article, Simkovicova, et al. [49] tested the genetic resistance of tomato against *Fusarium* and found that the accumulation of *PR2* (1-3)-β-glucanase and glucan endo-1,3-β-D-glucosidases limited *Fusarium* colonization.

WRKY transcription factors (TFs) are key regulators in signaling pathways in plant defense [50]. They participate in SA biosynthesis and the SA signaling network that includes the activation of *NPR1/3* and *PR1* [51]. In our system, at least three genes, *WRKY6*, *WRKY33*, and *WRKY70*, were upregulated in tested variants (Figure 10), which was consistent with the elevated levels of SA and the upregulation of *PR*- and *NPR1*-encoding genes. The results are in line with the findings that the ectopic expression of *WRKY6* in wheat improved broad-spectrum resistance to *Puccinia triticina* and *Fusarium* crown rot [52], and the overexpression of *WRKY33* resulted in improved immunity against two necrotrophic fungal pathogens in an SA-dependent pattern [53]. The third protein, WRKY70, functioned as a coordinator of the SA and JA signaling pathways. Its expression, regulated by *NPR1*, improved plant resistance through the SA-induced pathogenesis-related (PR) proteins [37]. As discussed above, the results of the transcriptome analysis revealed a complex pattern of changes and coordinated regulation in selected groups of barley genes responding to CS_10 treatment and *Fg* inoculation.

## 4. Materials and Methods

Materials. CS samples are specified in Table 1. All CS samples were supplied by Pol-Aura (Warsaw, Poland).

Acetic acid (99.5–99.9%) and sodium hydroxide (NaOH) were from POCH (Warsaw, Poland). Potato Dextrose Broth (PDB) was supplied by ROTH (Karlsruhe, Germany). Hoagland medium (Hoagland Modified Basal Salt Mixture) was provided by PhytoTech LABS (Lenexa, KS, USA). Salicylic acid (SA) was provided by Sigma-Aldrich (Saint Louis, MO, USA). Salicylic acid-d4 (SA-d4) was from Toronto Research Chemicals Inc. (North York, ON, Canada). 3,3-diaminobenzidine tetrahydrochloride (DAB) supplied from Pol-Aura (Warsaw, Poland).

Barley cultivation. Barley seeds, *Hordeum vulgare* cv Golden Promise, were imbibed for 24 h (4 °C in the dark) on Petri plates with glass beads and tap water followed by germination (48 h, 21 °C, in the dark) and planted in pots with substrate soil (Aura-Hollas^®^, Paslek, Poland) for all inoculation experiments. For biomass measurements, plants were grown in a semi-hydroponic system. Briefly, germinated seeds were placed on sterilized filter paper (20 cm × 50 cm) soaked in Hoagland medium, covered with another filter paper, and, after rolling up, placed in the glass jars with a 2 cm layer of Hoagland medium [54]. Plants in soil or semi-hydroponics were grown at 21 °C, 70–95% relative humidity, and a 16 h photoperiod with an illumination intensity of 250–300 µmol m^−2^ s^−1^.

The CS mock solution 4000 ppm was prepared by stirring 0.4 g of CS in 100 mL of 1% acetic acid (pH 3.0) overnight. The final solution, pH 5.6 (NaOH), was sterilized with a 0.22 µm pore diameter filter and stored at room temperature. The working CS 200 ppm solution was used for all plant treatments. The CS mock solution contained 0.05% acetic acid in water, pH 5.6 (NaOH). The second leaves of 14-day-old plants grown in soil were treated with CS solution or CS mock solution using a soft brush in experiments where CS-treated plants were used for further inoculation of third leaves with *Fg* and *Ph*. In experiments for biomass measurements, the upper part of the plants grown in semi-hydroponics were sprayed with either a CS solution (200 ppm) or CS mock solution.

*Fg* and *Ph* inoculation. *Fg* macroconidia were prepared as described [4]. Briefly, the V8-adapted liquid medium was inoculated with *Fg* mycelium, isolate BW5 (collection of Plant Breeding and Acclimatization Institute, Radzikow, Poland). After 2 weeks of culture (25 °C, continuous UV light (λmax = 365 nm), shaker 20 rpm), macroconidia were filtered through a sterile Miracloth. Suspension at a density of 10^6^ conidia mL^−1^ was aliquoted and stored at −80 °C until further use. For plant inoculation, 1 mL of *Fg* macroconidia suspension in ddH_2_O at a final density of 10^5^ conidia·mL^−1^ was infiltrated into barley third leaf with a syringe. The plants were kept in the dark for 24 h and further cultivated before infiltration. The control plants were treated with ddH_2_O. After inoculation, the plants were covered with a cellophane dome to ensure high humidity and kept in the dark for 24 h at 20 °C. After the cellophane dome was removed, the plants were treated with CS_10 200 ppm solution or mock solution (0.05% acetic acid). The inoculated plants were further cultured before additional inoculation. Inoculation with *Ph* urediniospores is based on the procedure outlined in [17]. Briefly, *Ph* urediniospores were suspended in mineral oil (Novec™ 7100 fluid) at a density of 0.5^−1^ mg spores·mL^−1^. The third leaves were inoculated by spraying suspended spores to obtain 40–150 spores cm^−2^ of the leaf surface. The inoculated plants were grown under 100% humidity for 24 h at 20 °C in the dark followed by cultivation under the same growth conditions as before inoculation.

To analyze barley immune response, the second barley leaves were sprayed with the mock or CS_10 solution, and the third leaves of the same plants were inoculated with macrospores of *Fg* or with *Ph* spores. The spatially separated CS_10 treatment (second leaf of the plant) and pathogen inoculation (the third leaf) allowed for the detection of plant immune response of the plant, not the direct antifungal effect of CS_10. The symptoms of pathogenesis were evaluated 5 days after inoculation with *Fg* and six days after inoculation with *Ph*.

DAB (3,3-diaminobenzidine tetrahydrochloride) staining. To evaluate the *Ph*–barley interaction, the third leaves, harvested 1, 2, 3, 4, 5, and 6 days post-inoculation (dpi) were stained with DAB. The leaf samples were placed in the DAB water solution (1 mg·mL^−1^, pH 3.8 in the dark) for 4 h and destained overnight in the mixture of ethanol–chloroform (3:1 *v*/*v*) with 0.15% trichloroacetic acid (TCA). The brown precipitate formed by the reaction of DAB with H_2_O_2_ was examined under a light microscope.

Calcofluor white (CW) staining. The third leaves inoculated with *Ph* spores were harvested and placed overnight in the mixture of ethanol–chloroform (3:1) with 0.15% TCA. After this, the samples were washed twice with 50% ethanol, 15 min each, twice with 0.05 M NaOH, 15 min each, and three times with water. Then, the samples were stained with CW 35 µg·mL^−1^ in 0.1 M Tris-HCl (pH 9.0, in the dark), washed once with 0.1 M Tris-HCl (pH 8.5), once with water, and stored in 25% glycerol with 0.1% lactophenol. The samples, observed under a fluorescence microscope (Nikon Diaphot, Aizu, Japan, epifluorescence optics with excitation at 340–380 nm, barrier filter at 420 nm, and dichroic mirror at 400 nm), were scored for the number of infection sites, appressoria, haustoria mother cells (HMCs), and micronecrotic reactions. For each time point, three biological replicates containing one entire leaf were used, and the percentage was calculated by dividing the total infection sites per leaf by the number of different kinds of infection symptoms and multiplying by 100. For DAB staining, the oxidative burst on the infected leaf was observed under the fluorescence microscope.

Salicylic acid quantification. To determine SA in the analyzed leaf tissues, a modified extraction procedure described by Verberne, et al. [55] was used. The tissues, collected from the second and third leaves of 14-day-old soil grown plants were ground in liquid N2, weighed, transferred to a 2 mL Eppendorf tube, mixed with 1 mL of MetOH:H2O (9:1 *v*/*v*), and extracted using a Retsch MM400 mill (2 × 5 min, 50 Hz) (Retsch, Haan, Germany). The whole homogenate, transferred to a 15 mL Falcon tube, was supplemented with MetOH–H2O (9:1 *v*/*v*) to a final ratio of 100 µL per 100 mg of fresh weight and the SA internal standard, SA-d4 at a final concentration of 100 µg per 100 mg of fresh weight. After extraction, the homogenate was filtered with a nylon membrane (pore diameter 0.45 µm) into Eppendorf tubes. The 2 mL of the filtrate was vacuum-dried under the stream of nitrogen in a water bath (60 °C). The dried samples, resuspended in 300 µL of 8M hydrochloric acid, were heated in closed tubes in a heating block at 80 °C for 1 h. As a result of hydrolysis, bound forms of SA were released, and they were extracted from the solution with a 1.5 mL mixture of ethyl acetate–cyclohexane (1:1 *v*/*v*). The extraction was repeated two more times in a similar manner. After each extraction, the upper layer was collected and combined in a 5 mL reaction vial. The solvent was evaporated under a stream of nitrogen in a heating block. The dried samples were dissolved in 0.5 mL of a mixture of methanol–water (9:1 *v*/*v*), sonicated, and filtered through a nylon syringe filter with a pore size of 0.22 µm and an outer diameter of 13 mm. These prepared filtered samples were analyzed using the LC-HRMS technique.

An ACQUITY H-Class high-performance liquid chromatography coupled to an LCT Premier XE high-resolution mass spectrometer (Waters, Milford, MA, USA) was used to analyze SA in the samples. Analytes were separated on a UPLC C18 Cortecs chromatography column (2.1 × 100 mm, 1.6 µm; Waters, Milford, MA, USA). The mobile phases consisted of methanol and water, i.e., 10:90 *v*/*v* (phase A) and 90:10 *v*/*v* (phase B). Both phases contained 0.1% formic acid and 5 mM ammonium formate. The flow rate was 0.3 mL/min. The following gradients were used: 100% phase A from 0 to 2 min; 50% phase A from 3 to 6 min; 10% phase A from 10 to 13 min; 100% phase A from 14 to 16 min. Five microliters of the sample were injected onto the column. The mass spectrometer was operated in negative polarization with electrospray ionization (ESI). The ion source and desolvation temperatures were 125 and 370 °C, respectively. The flow rate of the spray gas (nitrogen) was 650 L/min, and the flow rate of the drying gas was 20 L/min. The voltage on the capillary was 2200 V. Mode V of ion optics was used. The mass spectrometer was calibrated using a standard leucine-enkephalin solution. Test compounds (SA and SA-d4) in barley leaves were identified by comparing their molecular weights and retention times with standards of these substances. The following molecular ions (m/z) were used for quantitative analysis: SA–137.110 (M-H)-; SA-d4–141.113 (M-H)-.

To determine the reliability of the method, a validation experiment was performed to determine the recovery (R) and repeatability of the method (expressed as the mean standard deviation RSD). The limit of quantification (LOQ) and the limit of detection (LOD) were determined, as well as the linearity range. The recovery of the method was determined by adding the SA standard to the leaf tissue sample at three levels of addition, that is, 1, 2, and 3 mg/kg, and the addition of SA-d4 at the level described above. SA concentrations were determined using internal calibration. The calibration method was based on isotopic dilutions using SA-containing stable isotopes. The extraction procedure was used as described in the previous section. Recovery was determined from the difference in SA content in the fortified and blank samples used in the experiment. The average recovery was determined from at least three independent determinations. To determine the LOQ value, the SA concentration was assumed, at which the signal-to-noise ratio was at least 10 and, in the case of LOD, at least 3. Calibration solutions were prepared at seven different concentrations ranging from 0.025 to 1.6 mg/kg. SA-d4 was added to each solution in an amount corresponding to SA-d4 in the tested samples. The conducted validation experiment allowed us to determine the effectiveness of the analytical method. The recovery obtained for SA was 97% at the 1 mg/kg level: 93% for 2 mg/kg and 101% for 3 mg/kg. The repeatability of the method (RSD) was 7, 11, and 6%, respectively, while LOQ and LOD were 0.100 and 0.03 mg/kg. The value of the determination coefficient (R^2^) for the determined calibration curve is at least 0.99.

RNA extraction and mRNA sequencing. Total RNA was extracted from 100 mg of barley leaves representing four tested variants, *Fv*_mock, *Hv*_CS, *Hv*_*Fg*_CS, and *Hv*_*Fg* (Figure 12), using an RNA isolation kit (Zymo-R2072, Irvine, CA, USA), RNA extraction buffer (Tris-HCl 50 mM pH = 8.0, LiCl 150 mM, EDTA 5 mM pH = 8.0, SDS 1%), Trizol reagent, phenol–chloroform (1:1 *v*/*v*) mixture, and 80% ethanol, according to the manufacturer’s protocol. The isolated RNA was quantified using a NanoDrop spectrophotometer (NanoDrop Technologies^®^, Wilmington, DE, USA). The quality of RNA preparations was assessed on BioAnalyzer 2100 using 1% agarose gel (Agilent Technologies^®^, Santa Clara, CA, USA). Samples with a ration A260/A280 over 2.0, RIN number over 8, and concentration over 50 ng µL^−1^ were selected for downstream applications. Approximately 2 µg of high-quality RNA was sent to a biotech company (GENEWIZ^®^, Leipzig, Germany) to prepare the Illumina standard RNA library with polyA selection. The company performed Illumina NovaSeq sequencing with an estimated data output of ~20 M paired-end reads with a quality score of Q30. The company provided FASTQ format data that were used for further analysis.

The quality control and mapping sequence reads. FastQC toolkit version 0.12.0 (https://www.bioinformatics.babraham.ac.uk/projects/fastqc, accessed on 15 June 2024) was used to perform a quality check on the FastQ raw data, and default commands were used [56]. For alignment, the reference genome of Hordeum vulgare L. Golden Promise (Assembly: GCA_902500625.1, obtained from https://www.ebi.ac.uk/ena/browser/home, accessed on 15 June 2024) was used in Bowtie version 2.5.0 for the alignment of previously trimmed reads [57]. SAM tools were used to convert output alignment files into .bam files. Converted files were used for transcript reads analysis with the use of FeatureCounts and further analyzed with DESeq2 for differentially expressed genes. The cut-off values for DEG (differentially expressed genes) were false discovery rate (FDR) < 0.05 and log2-fold change > 2.

Analyzing RNA-Seq outputs. To determine variation between samples, principal component analysis (PCA) Bioconductor R PGSEA package (https://www.bioconductor.org/packages//2.10/bioc/html/PGSEA.html, accessed on 21 June 2024) with default settings was used. Hierarchical clustering heatmap and correlation matrix of biological replicates of each treatment were prepared using iDEP web application (http://bioinformatics.sdstate.edu/idep/, accessed on 21 June 2024). For functional enrichment analysis, the iDEP web application was used to interpret previously processed data with a false discovery rate (FDR) cutoff of <0.05 and a log2-fold change > 2.0. Gene Ontology (GO) terms were used to analyze pathways that underwent significant changes in their regulation based on biological processes (BPs), molecular functions (MFs), and cellular components (CCs) [58].

RT-qPCR of selected *H. vulgare* genes. Total RNA was extracted as stated in RNA extraction iboLock RNase inhibitor and 2 U of DNase1 (Roche Diagnostics^®^, Munich, Germany). Complete DNA degradation was confirmed by using 100 ng DNase-treated RNA as a template in a PCR reaction with *ARF* (*ADP-Ribosylation factor*, AJ508228) gene primers. Extracted RNA was used as the template for cDNA synthesis using the Revert Aid First Strand cDNA Synthesis Kit (ThermoFisher Scientific^®^, Waltham, MA, USA) along the manufacturers protocol (with use of oligo-dt primers). Obtained cDNA, 5-fold diluted, was used as a template for quantitative PCR. The qPCR reaction mix contained 2 μL of 5× HOT FIREPol EvaGreen qPCR Mix Plus (ROX) (Solis Biodyne^®^, Tartu, Estonia), 0.3 μL of primer F (10 μM), 0.3 μL of primer R (10 μM), 15 ng of template cDNA, and 10 μL of water. BioRad CFX384 Real-Time PCR System (BioRad, Hercules, CA, USA) was used for qPCR reaction The relative expression of selected barley genes was based on the 2^−∆∆Ct^ method, as previously described [59]. Values for the relative number of transcripts represent a medium with a minimum of three biological replicates and three technical replicates. The geometric mean of two housekeeping genes AJ508228 [56] and AK362208.1 [57] was used as reference for RT-qPCR of selected barley genes. All the primers used for qPCR are listed in Appendix A.

*Fg* genomic DNA quantification by qPCR. The inoculated third leaves of the CS_10- and mock-treated plants were harvested and ground in liquid N2. Genomic DNA (gDNA) was extracted from the leaves using the CTAB method. Briefly, 0.5 g of the N_2_ ground sample was extracted with 800 µL of CTAB solution containing β-ME (2 µL mL^−1^ CTAB). The samples were incubated at 60 °C for 30–40 min, followed by extraction with 800 µL of chloroform–isoamyl alcohol (24:1 *v*/*v*) and centrifugation (10,000 rpm, 20 min) at room temperature. Subsequently, 600 µL of the supernatant was transferred into a new Eppendorf tube supplemented with 5 µL of RNase and incubated for 10–15 min at 37 °C. gDNA was precipitated by adding isopropanol (3/4 of the supernatant volume), freezing for 30 min, and centrifuging for 10,000 rpm, 10 min at room temperature. The precipitate was washed with 70% ethanol, and the pellet was dissolved in 200 µL of the TE buffer. The quality and quantity of DNA were checked using a Nanodrop, followed by 1% agarose gel electrophoresis. *Fusarium* DNA was quantified in total DNA isolated from infected leaves using qPCR, total DNA as a template and primers to barley *Hv_EFG1* (*translation elongation factor G1*, AY836205.1 [56]), and *Fg* gene *Fg_TRI5* (*trichodiene synthase5*, [60]). The qPCR reaction mix consisted of 2 μL of HOT FIREPol EvaGreen qPCR Mix Plus (ROX) (Solis Biodyne^®^, Tartu, Estonia), 50 ng of gDNA extracted from the respective sample, 0.3 μL of forward primer (10 μM), and 0.3 μL of reverse primer (10 μM) for each target gene. The quantification was based on three biological replicates and four technical replicates for each sample. The results were shown as the relative number of *Fg_TRI5* gene copies per *Hv_EFG1* gene copy. The primers used for qPCR are listed in Appendix A.

Measurement of barley biomass. The leaves of semi-hydroponically grown barley seedlings were subjected to two separate applications of CS at a concentration of 200 ppm, the first one 3 days after imbibition and the second one 10 days after imbibition. The biomass was weighed twice for each plant after careful removal from the paper filter (Figure 13).

The effects of CS and SA on plant growth were evaluated in two independent experiments. To account for biological variation among individual plants, biomass gain data were collected for individual plants from five independent experiments involving over 300 plants treated with CS and five experiments with over 300 plants treated with SA. Biomass measurements were recorded for 5-day-old (M5) and 24-day-old (M24) seedlings. These measurements were used to calculate the individual biomass gain (BG) for each seedling across the tested variants using Equation (1).
BG = M24 − M9(1)

To compare the biomass gains (BGs) of seedlings treated with CS and SA, the relative biomass gain (RBG) was calculated. For mock-treated plants, the RBG was set as 1.0. The RBG values for both treatments (CS and SA) were determined using Equation (2).
RBG = BG_treatment_/BG_mock_(2)

The resulting data were then analyzed using one-way ANOVA, followed by Tukey’s post hoc test for statistical comparisons.

Statistical analysis. All quantitative data are presented as the mean value and standard error. The data were processed by one-way analysis of variance (ANOVA) and post hoc least significant difference (LSD) or Tukey’s test using Statistica 13 (StatSoft Polska, Kraków, Poland). Statistically significant results were marked with the following values: * *p* < 0.05, ** *p* < 0.01, and *** *p* < 0.001.

## 5. Conclusions

The batch of CS_10 with the strongest antifungal activity against *Fg* was also effective in activating plant immune responses and enhancing the growth of plant biomass.The immune response induced by the application of CS_10 led to increased resistance against two pathogenic fungi, *Ph* and *Fg*, representing biotrophic and necrotrophic types of pathogenesis, respectively.The two observed effects of CS_10 treatment in plants, i.e., enhanced immunity and biomass growth enhancement, were mediated by SA-dependent regulation, highlighting the dual role of CS in plant immunity and growth regulation. Barley transcriptome analysis confirmed that the activation of the immune response involved SA-regulated genes.

## Figures and Tables

**Figure 1 ijms-25-13244-f001:**
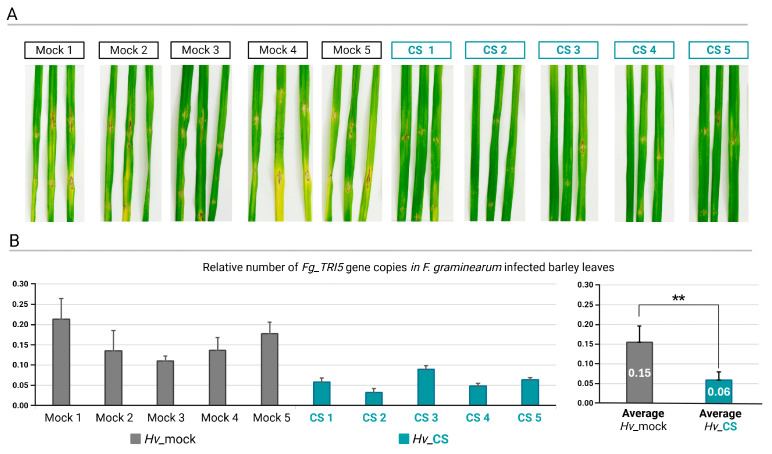
Representative picture of infection symptoms on the third barley leaves inoculated with *Fusarium graminearum* (*Fg*) in plants where the second leaves were mock (*Hv*-mock)- or chitosan_10 (CS)-treated (*Hv*_CS) (**A**). Relative number of *Fg TRI4* gene copies (*Fg_TRI5*) per one copy of barley *EFG1* gene (*Hv_EFG1*) is shown. The results are from five independent biological repetitions and the average values of genes’ quantification are shown (**B**). Asterisks indicate significance level (based on one-way ANOVA and Tukey’s post hoc test) ** *p* ≤ 0.01.

**Figure 2 ijms-25-13244-f002:**
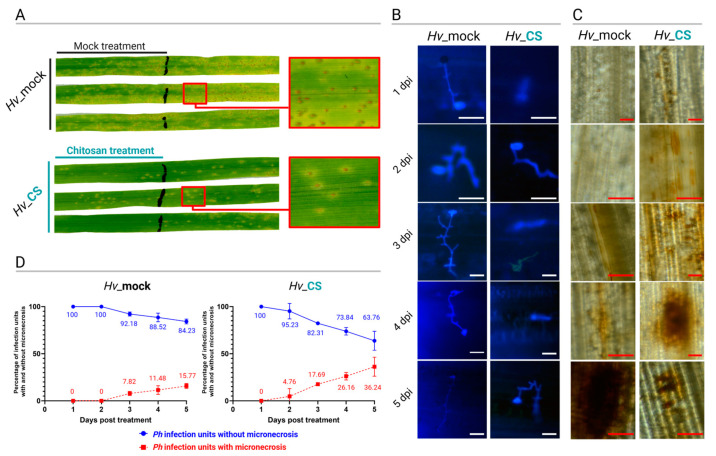
Plant–pathogen interaction of barley plants treated with mock (*Hv*-mock) or with chitosan_10 solution (*Hv*-CS) followed by inoculation with *Puccinia hordei* (*Ph*) urediniospores. The CS_10 or mock treatments were applied to the second leaves of the plants, and the third leaves of the same plants were inoculated with *Ph* urediniospores. This approach allowed us to detect the results of plant immune response induced by the CS-10 and not a direct inhibitory effect of the CS-10 on the pathogen. Representative pictures of infection symptoms on the third leaves of mock- and CS_10-treated plants scored six days post-inoculation (**A**). Representative pictures of microscopic observation of infection sites of calcofluor white stained leaf samples scored from 1 to 5 days post-inoculation. Scale bars = 100 µm (**B**). Representative pictures of leaf samples stained with DAB. Scale bars = 100 µm (**C**). The rates of micronecrotic reactions in *Ph* infection units on barley leaves. The mean values and standard deviation were calculated based on scoring one entire leaf from each time point and three biological replicates (**D**).

**Figure 3 ijms-25-13244-f003:**
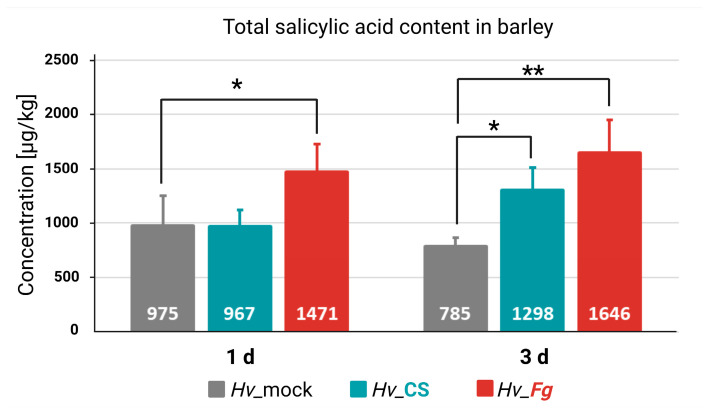
Concentration of total salicylic acid (SA) in barley leaves collected one day (1 d) and three days (3 d) after mock (*Hv*-mock) or chitosan_10 treatment (*Hv*-CS), or inoculation with *F. graminearum* (*Fg*) (*Hv*_*Fg*). Asterisks indicate significance level (based on one-way ANOVA and LSD post hoc test) * *p* ≤ 0.05 and ** *p* ≤ 0.01.

**Figure 4 ijms-25-13244-f004:**
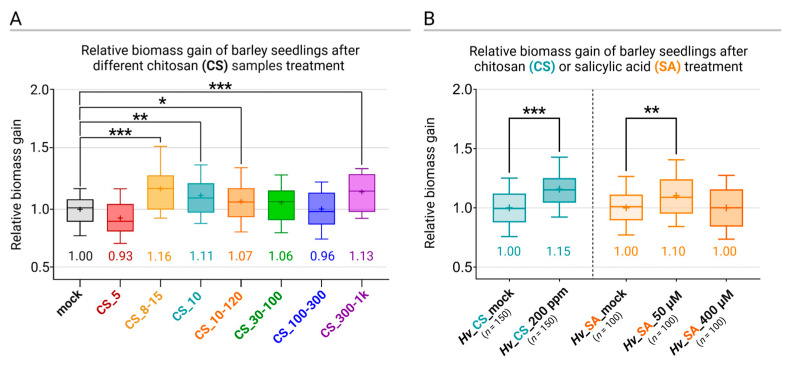
Relative biomass gain of barley seedlings after 19 days of cultivation in Hoagland medium after treatment with seven chitosan batches (200 ppm): CS_5, CS_8-15, CS_10, CS_10-120, CS_30-100, CS_100-300, and CS_300-1000. For each sample, 40 separate plants have been tested (**A**). Relative biomass gain of barley seedlings after 19 days of cultivation in Hoagland medium after chitosan (CS_10, 200 ppm) and after salicylic acid (SA, 50 μM and 400 μM) treatment (**B**). Each box represents the percentile in range 25–75; the whiskers represent the 10 and 90 percentiles. Asterisks indicate significance level (based on one-way ANOVA and Tukey’s post hoc test) * *p* ≤ 0.05, ** *p* ≤ 0.01, and *** *p* ≤ 0.001.

**Figure 5 ijms-25-13244-f005:**
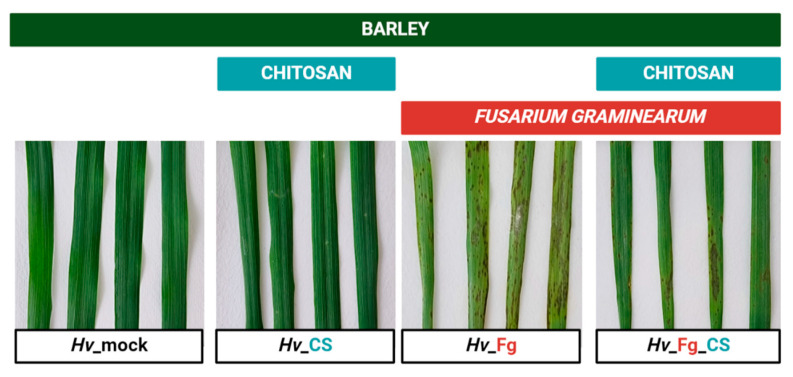
Representative picture of leaf samples used for RNA-seq analysis. *Hv*_mock—leaves treated with mock solution containing 0.05% acetic acid; *Hv*_CS—leaves treated with CS_10 (solutions of CS_10 also contained 0.05% acetic acid); *Hv*_*Fg*—leaves inoculated with *F. graminearum* (*Fg*); *Hv*_*Fg*_CS—leaves inoculated with *Fg* and treated with CS.

**Figure 6 ijms-25-13244-f006:**
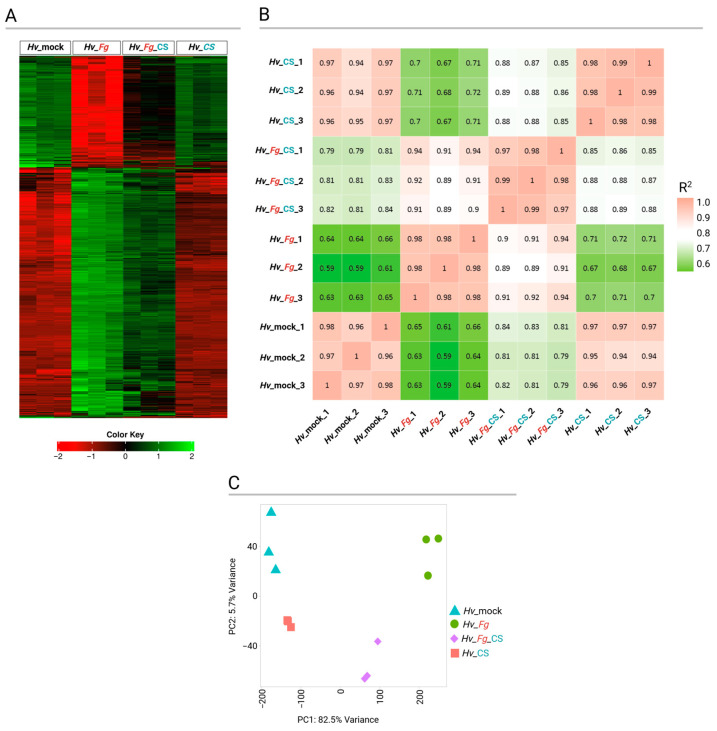
Hierarchical clustering heatmap of tested variants: leaf control samples (*Hv*_mock), leaves treated with CS_10 (*Hv*_CS), inoculated with *F.* graminearum (*Fg*) (*Hv*_*Fg*), and treated with CS_10 and inoculated with *Fg* (*Hv*_*Fg*_CS). The three columns in each variant represent the three biological replicates (**A**). Correlation matrix of all three biological replicates of each tested variant (**B**). Principal component analysis of all tested variants (**C**).

**Figure 7 ijms-25-13244-f007:**
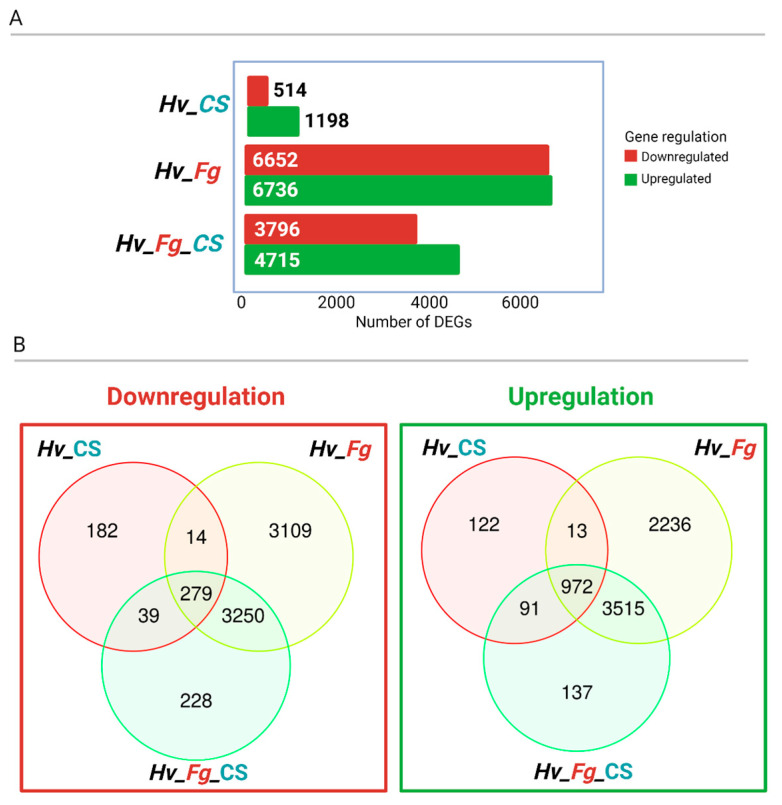
Numbers of differentially expressed genes (DEGs) in analyzed samples in relation to the control (*Hv*_mock). The tested variants include leaves treated with CS_10 (*Hv*-CS), leaves inoculated with *F. graminearum* (*Fg*) (*Hv*-*Fg*), and leaves treated with CS_10 and inoculated with *Fg* (*Hv*_*Fg*_CS) (**A**). Venn diagrams showing number of differentially expressed genes (DEGs) in each the three tested variants in relation to mock-treated control samples. Variants: leaves treated with CS_10 (*Hv*-CS), leaves inoculated with *Fg* (*Hv*-*Fg*), and leaves treated with CS_10 and inoculated with *Fg* (*Hv_Fg*_CS) (**B**). Presented genes are based on a cutoff value of FDR < 0.05 and log2fold change > 2.

**Figure 8 ijms-25-13244-f008:**
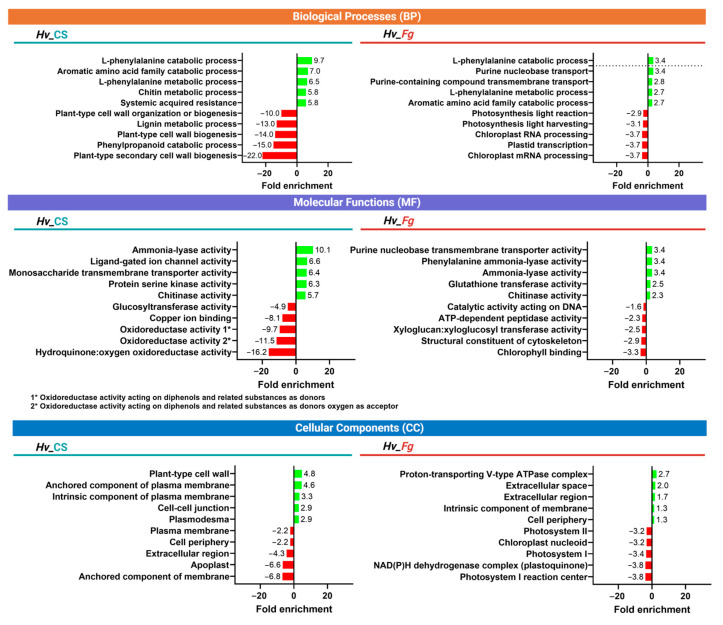
Top five Gene Ontology terms sorted by fold enrichment across chitosan_10 treated barley (*Hv*_CS) and *F. graminearum* inoculated barley (*Hv*_*Fg*) categorized into BPs (biological processes), MF (molecular function) and CC (cellular component) gene sets.

**Figure 9 ijms-25-13244-f009:**
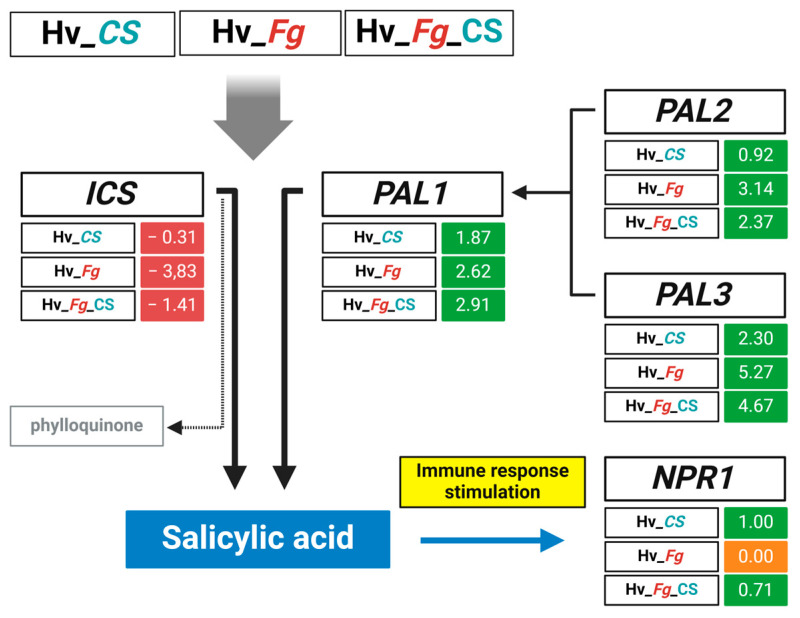
Regulation pattern of PAL- and ICS-encoding genes in variants of chitosan_10-treated (*Hv*_CS), *F. graminearum* (*Fg*)-inoculated (*Hv_Fg*), and CS_10-treated and *Fg*-inoculated barley.

**Figure 10 ijms-25-13244-f010:**
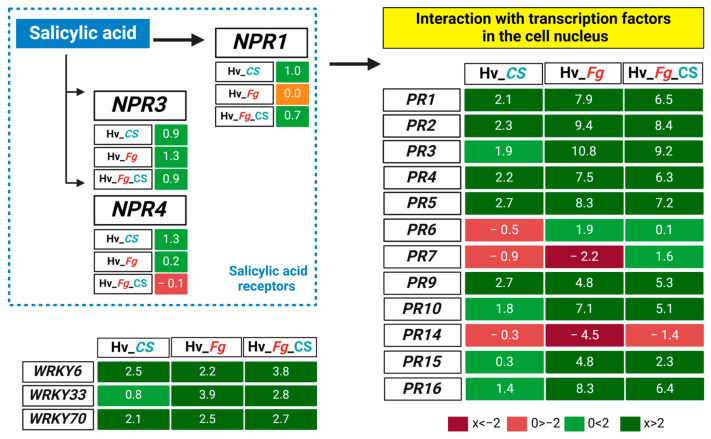
Regulation pattern of genes encoding NPR1, NPR3, and NPR4 regulators, selected WRKY transcription factors and pathogenesis-related (PR) proteins in variants of chitosan_10-treated (*Hv*_CS), *F. graminearum* (*Fg*)-inoculated (*Hv_Fg*), and chitosan_10-treated and *Fg*-inoculated barley plants. The blue color indicates the SA-related genes and pathways.

**Figure 11 ijms-25-13244-f011:**
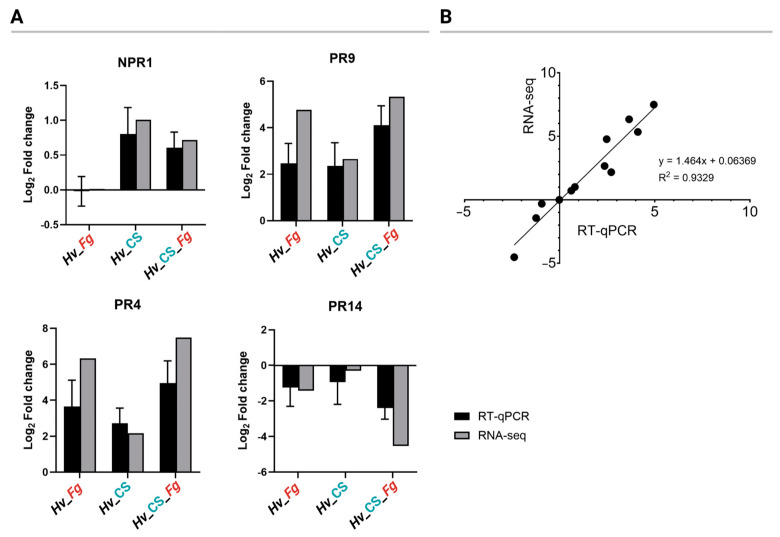
Validation of RNA-seq differentially expressed genes (DEGs) using RT-qPCR of four genes *NPR1*, *PR9*, *PR4*, and *PR14*. The log2-fold change values (**A**) and the linear regression between the log2-fold change of RNA-seq and RT-qPCR quantification are shown. The points represent individual results for each gene and the three variants (**B**).

**Figure 12 ijms-25-13244-f012:**
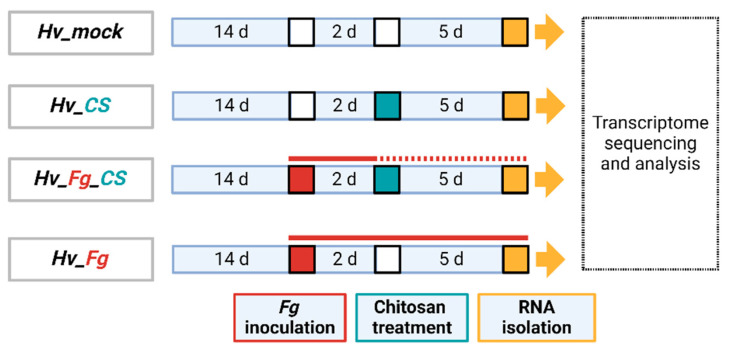
Schematic representation of experimental variants for transcriptome sequencing. Briefly, 14-day-old plants were inoculated with *F. graminearum* (*Fg*), followed by chitosan_10 (CS) treatment two days later and a collection of samples 5 days later. Description of tested variants: *Hv*_mock—barley treated with mock solution; *Hv*_CS—barley treated with chitosan 200 ppm; *Hv_Fg*_CS—barley inoculated with *Fg* and treated with chitosan 200 ppm; and *Hv_Fg*—barley inoculated with *Fg*.

**Figure 13 ijms-25-13244-f013:**
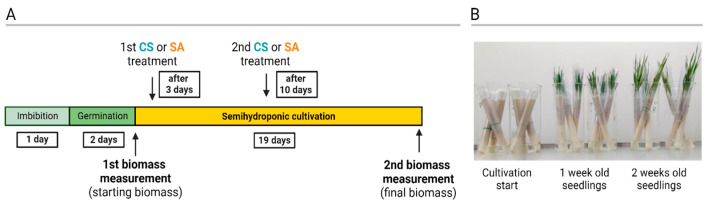
Schematic timeline of biomass measurements and chitosan (CS) or salicylic acid (SA) treatments (**A**). Representative picture of barley plants grown in semi-hydroponics (**B**).

**Table 1 ijms-25-13244-t001:** Characteristics of the selected chitosan (CS) samples used in the experiments.

No.	Name of the Sample	Viscosity [cps]	Molecular Weight [kDa]	Deacetylation Degree [%]	Origin
1	CS_5	5	20	≥90	Shrimp
2	CS_10	10	30	≥90	Shrimp
3	CS_8-15	8–15	20–100	87.6–92.5	Shrimp
4	CS_10-120	10–120	NP *	≥85	*Aspergillus niger*
5	CS_30-100	30–100	250	≥90	Shrimp
6	CS_100-300	100–300	890	≥90	Shrimp
7	CS_300-1k	300–1000	1250	≥90	Shrimp

* NP—characteristic not provided by the distributor.

## Data Availability

All data and materials are included in the article and the Appendix A. Raw data sets supporting the results of this article are available in the NCBI Sequence Read Archive repository: Accession: PRJNA1170567 ID: 1170567 ‘Effect of Chitosan on the Transcriptome of the *Fusarium graminearum* and *Hordeum vulgare* L. Pathosystem’.

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
