# Peer review of "Salicylic Acid Mediates Chitosan-Induced Immune Responses and Growth Enhancement in Barley"

_ijms, 2024, doi:10.3390/ijms252413244_

Round 1
Reviewer 1 Report
Comments and Suggestions for Authors
Manuscript Number: ijms-3357000
Title: Salicylic Acid Mediates Chitosan-Induced Immune Responses and Growth Enhancement in Barley
The authors' previous study found that CS with different molecular weights had different antifungal activities. Based on this, it was found that CS with high antifungal activity could effectively activate plant immune response and promote biomass growth. Barley treated with this CS exhibited systemic acquired resistance and enhanced defense against Fusarium graminearum and Puccinia hordei. In addition, CS treatment promoted salicylic acid (SA) accumulation. The effect of exogenous SA application was similar to that of the CS treatment, suggesting that SA is key to the CS-induced response.
However, this research article is notable for the improvements that could the English could be improved to more clearly express the research. Therefore, the manuscript needs to minor revise for possible publish in International Journal of Biological Macromolecules.
The comments are provided below:
1. References need to be inserted in lines 29-33.
2. In lines 33, 131 et seq., replace “Chitosan (CS)” with “CS”.
3. Line 45, change “enhancement of plant immunity position CS as a promising compound for” to “enhancement of plant immunity position,CS as a promising compound for”.
4. In lines 52, 130 et seq., replace “Fusarium graminearum (Fg)” with “Fg”.
5. Line 104, change “F. graminearum and P. hordei” to “Fg and Ph”.
6. Lines 65-66,change “reactive oxygen species (ROS)”to“ROS”.
7. There are multiple instances of repeated labeling of abbreviations in the article, where subsequent content can be used directly after the first occurrence, and it is recommended that this be checked in full.
8. Lines 138-144, the description of which has been mentioned earlier, are suggested to be simplified to increase readability.
9. In lines 156-157, replace “The infection rate of micronecroses at 3 dpi was 7,8% and it increased to 11,5 % and 15,8 % at 4 and 5 dpi, respectively.” with “The infection rate of micronecroses at 3 dpi was 7.8% and it increased to 11.5 % and 15.8 % at 4 and 5 dpi, respectively.”.
10. Line 164, change “A higher accumulation of H2O2” to “A higher accumulation of H2O2”。
11. Lines 222-237 of experimental descriptive content suggests that it be placed in the Materials and Methods.
12. Line 294, Hv needs to be italicized.
Comments on the Quality of English Language
The English could be improved to more clearly express the research.
Reviewer 2 Report
Comments and Suggestions for Authors
The authors of the manuscript titled “Salicylic Acid Mediates Chitosan-Induced Immune Responses and Growth Enhancement in Barley” investigated the effects of chitosan application on barley growth and its immune response to fungal pathogens. They identified the factors contributing to an enhanced immune response and resistance and increased plant biomass growth. The results indicated that applying chitosan to barley plants improved both their immunity and biomass growth. These enhancements were facilitated by the regulation dependent on salicylic acid (SA), which underscores chitosan's dual role in promoting plant immunity and growth. Additionally, an analysis of the barley transcriptome confirmed that the activation of the immune response was linked to genes regulated by SA.
Specific comments:
Abstract:
L13 and 14: ‘CS batch’ and ‘this CS’ Please indicate the chitosan for better understanding.
L17: ‘CS treatment’ Please indicate the concentration of chitosan.
L18: Please provide the fold change or percentage increase in plant biomass growth.
Keywords: ‘Barley’ Please delete it.
Introduction: Add information about the characteristics of the barley cultivar used in this study.
L41: Please provide more details about ‘plant immune responses and stimulate growth’.
L94: ‘batch of CS,’ please indicate the chitosan and its concentration.
L98: ‘plant’ barley?
L99: ‘Hordeum vulgare cv Golden Promise’ instead of ‘barley’s’
Results: Please avoid the methods in the results. For example: L110-115.
L143: Which is correct: "seven days post-inoculation" or "six days post-inoculation" as shown in Figure 2A?
L164: H2O2 (typo). There are several typographical errors in the manuscript, primarily in the methods section. Please correct them.
L189-196: Please rewrite the sentences.
L211: Please indicate the antifungal activities of CS_8-15 and CS_10-120 (supplementary file).
L215: CS_10 is referred to simply as 'CS'. Please define the abbreviation at the first use: CS_10 (CS). For chitosan, do not use the abbreviation to avoid confusion; use chitosan instead of chitosan (CS).
L243: What is the difference between Figure 4A and Figure S1?
Discussion: Please avoid subheadings.
L92-96: Repetition of introduction and results.
L433: Fusarium zanthoxyli (Italic). Please italicize the species name throughout the text, including in the references.
Materials and Methods: Please correct the typos.
L570: ‘CS solution’ The authors conducted tests on all the chitosan samples listed in Table 1. Please provide the data (2.1).
L577: Please indicate the properties of UV light.
Author Response
Please see the attechement.
